# Arthropod Diversity Influenced by Two *Musa*-Based Agroecosystems in Ecuador

**Daniel Vera-Aviles [1],\* , Carmita Suarez-Capello [1], Mercè Llugany [2], Charlotte Poschenrieder [2] , Paola De Santis [3,4] and Milton Cabezas-Guerrero [5]**

[1] Facultad de Ciencias Agrarias, Universidad Técnica Estatal de Quevedo, Quevedo 120509, Ecuador; csuarez@uteq.edu.ec

[2] Laboratory of Plant Physiology, Universitat Autònoma de Barcelona, Bellaterra, 08193 Barcelona, Spain; merce.llugany@uab.cat (M.L.); Charlotte.Poschenrieder@uab.cat (C.P.)

[3] Research Centre Bioversity International, Via dei Tre Denari 472/a, 00054 Maccarese (Fiumicino), Italy; p.desantis@cgiar.org

[4] Department Environmental Biology, Università Sapienza, Ple Aldo Moro, 5 00185 Rome, Italy

[5] Facultad de Ciencias Pecuarias. Universidad Técnica Estatal de Quevedo, Quevedo 120509, Ecuador; mcabezas@uteq.edu.ec

\* Correspondence: dvera@uteq.edu.ec

**Abstract:** Banana and plantain (*Musa* spp.) are very important crops in Ecuador. Agricultural production systems based on a single cultivar and high use of external inputs to increase yields may cause changes in the landscape structure and a loss in biodiversity. This loss may be responsible for a decrease in the complexity of arthropod food webs and, at the same time, related to a higher frequency and range of pest outbreaks. Very little is known either about the ecological mechanisms causing destabilization of these systems or the importance of the diversity of natural enemies to keep pests under control. Few studies have focused on this issue in tropical ecosystems. Here, we address this problem, comparing two *Musa*-based agroecosystems (monocultivar and mixed-species plantations) at two sites in Ecuador (La Maná and El Carmen) with different precipitation regimes. The diversity of soil macro fauna, represented by arthropods, was established, as indicators of the abovementioned disturbances. Our ultimate goal is the optimization of pest management by exploring more sustainable cropping systems with improved soil quality. Arthropod abundance was higher in the mixed system at both localities, which was clearly associated with the quality of the soils. In addition, we found Hymenoptera species with predatory or parasitic characteristics over the pests present in the agroecosystems under study. These highly beneficial species were more abundant at the locality of La Maná. The mixed type of production system provides plant diversity, which favors beneficial arthropod abundance and permits lower agrochemical application without yield penalties in comparison to the monoculture. These findings will help in the design of *Musa*-based agroecosystems to enhance pest control.

**Keywords:** monocultivar; mixed-species plantation; biodiversity; arthropod; soil; on-farm biodiversity indicators

## 1. Introduction

At present, biodiversity is being lost at an unprecedented rate due to human activities. Research has devoted a great deal of effort to assess the importance of biodiversity for the functioning and stability of agroecosystems and for the provision of environmental services. Pest management has become more efficient on numerous occasions, as a valuable environmental service provided by biodiversity. However, this is threatened by human activity [1,2].

It is well known that agricultural production systems are intensified by enhanced use of external inputs to increase yields, causing a change in the landscape structure. These intense cropping systems are prone to losing their biodiversity and become destabilized. Studies carried out by Michalko and Košulič [3] showed that components of biodiversity peaked at different levels of canopy openness on oak forest stands. Therefore, the restoration and suitable forest management of such conditions will retain important diversification of habitats. Diversity is an index composed of two variables: the abundance of species (or groups of species) and the equitability (or uniform distribution of individuals between groups) [4]. Michalko and Košulič [3] suggested that the permanent presence of small-scale improvements could be suitable conservation tools to prevent the general decline of woodland biodiversity in the intensified landscape, and proposed that all the suggested improvements would promote species diversity, conservation aspects, and functional diversity. A loss in plant diversity decreases the complexity of arthropod food webs that could be related to the observed higher frequency and range of pest outbreaks [5]. However, little is known about the ecological mechanisms that result in this destabilization or the importance of the diversity of the natural enemies to keep the pest under control [6]. In this context, it is essential to identify those species/orders/families that, with their presence or absence, are important indicators of the quality of the agroecosystem under study. The soil microbiota represents the largest group of terrestrial animal organisms; among them, the phylum Arthropoda is considered one of the most important for man; despite its small size, they are visible to the naked eye, have a relatively short life cycle and fulfill numerous environmental services [7].

In the last decade, the diversity of the different production systems has been a matter of concern due to changes in ecosystems by human activities in agriculture, livestock and forestry [8]. These actions cause a great alteration in the processes of configuring the habitat of the organisms occupying that environment, causing negative effects on the diversity of the macro fauna and altering the balance between the ecosystem, the soil and the plants [9]. We suppose that all the suggested improvements would promote species diversity, conservation aspects, and functional diversity. These are essential to ecosystem functions and the restoration of forest environments in landscapes under intense human land use.

The decrease in diversity and abundance of the edaphic macro fauna caused by human activity has generated concerns that have stimulated the development of research on the impact of different types of tillage on soil biota. A clear example is described by Arroyo and Iturrondobeitia [10], who evaluated the diversity of arthropods in forests and different monocultivar agricultural systems, finding high values of species richness in forest areas, in contrast with the agroecosystem receiving fertilization and a general management of the crop. These kinds of studies are very scarce in the tropics, especially those related to agricultural crops.

Banana and plantain monocultivar plantations in Ecuador have been established in areas where primary forest has been eroded. A feature of these tropical soils is their dependence on the biomass and species diversity of the forest that covers it. Once the protective cover of the forest is eliminated, the productivity and fertility per unit area decreases.

Banana monocultivar are affected by many pests due to their weak ecological balance [11]. This drastically reduces the yield after the first two cropping years. For this reason, the banana (*Musa balbisiana*) producers need large areas of land and the consequent expansion to compensate for the fall in production per hectare. Extensive commercial banana crops are governed by technological standards aimed towards the intense production of fruit for export. Plantains (*Musa acuminata*), instead, besides being an important export crop, are also an essential staple food of coastal Ecuador and many other countries along the tropics, where diversity in respect to pest and disease pressures exists. Although farmers try to apply basically the same management practices as banana enterprises, plantain is in the hands of medium and small farmers with limited access to resources and who use lower levels of technology.

There are different approaches to tackle those problems. One approach intensively used is integrated pest management (IPM) strategies, focusing on using agronomic management techniques to reduce pesticide use, but IPM concentrates on modifying the environment around predominantly

modern cultivars and has tended to exclude the potential of using within-crop diversity through genetic mixtures (crop variety mixtures) for example, or the planned deployment of different varieties in the same production environment. A diverse genetic basis of resistance (e.g., crop variety mixtures) is beneficial for the farmer because it allows a more stable management of pest and disease pressure than what a monocultivar system allows.

Intraspecific biodiversity increases resistance to pests in crops, developing more biological support, thus, ensuring production. Greater intraspecific biodiversity improves the biota in the soil, creating synergy with the crop. The mixture of natural and human selection gives the particularity of the environment for different agroecosystems. It requires agricultural species with genetic characteristics that adapt to the different environments. The most palpable case is that of maize, conducted in different climatic zones under different constrains [12].

There is very little information and research work of this kind in the region, therefore, the development of a project in progress under an agreement between the Technical University of Quevedo, Ecuador and Bioversity International [13] was used to conduct this study in order to determine possible bioindicators of the use of intraspecific biodiversity in two banana production systems on the Ecuadorian coast. The present work aimed at developing a study of the *Arthropoda* population over two Musa agroecosystems (single and mixed Musa cultivar plantations, at two sites in Ecuador (La Maná and El Carmen)), to establish its value as a bioindicator under Ecuadorian conditions.

## 2. Materials and Methods

### 2.1. Site Description

The study was conducted at two sites in Ecuador with different soil characteristics and precipitation regimes (Table 1) in 2014. La Maná site is located in the province of Cotopaxi (00°53′43″ S; 79°11′05″ O) at 354 m altitude. El Carmen site is located in Manabí province (00°16′14″ S; 79°29′12″ O) at 250 m altitude.

**Table 1.** Soil and weather characteristics of both sites and agricultural systems studied during 2014.

| Soil Characteristic | La Maná | | EL Carmen | |
| --- | --- | --- | --- | --- |
| | **Mixed** | **Monocultivar** | **Mixed** | **Monocultivar** |
| Texture | Sandy loam | Sandy loam | Silty loam | Silty loam |
| pH | 5.9 | 5.5 | 6.4 | 6.0 |
| Organic matter (%) | 4.3 | 2.9 | 4.0 | 3.2 |
| $NH_4$ (ppm) | 26 | 24 | 27 | 24 |
| **Weather Variables** [1] | **Rainy Season** | **Dry Season** | **Rainy Season** | **Dry Season** |
| Precipitation (mm) | 2589 | 398 | 2084 | 461 |
| Temperature (°C)     max | 29.2 | 27.5 | 30.3 | 28.7 |
| min | 20.4 | 19.4 | 20.4 | 19.2 |
| med | 24.8 | 23.4 | 25.3 | 23.9 |
| RH (%) | 89.2 | 87.3 | 87.6 | 86.9 |
| Light hours | 231 | 383 | 328 | 447 |

[1] Source: National Institute of Meteorology and Hydrology (INAMHI), San Juan Station (La Maná) and El Carmen Station (Manabí).

### 2.2. Experimental Design

The experimental sites consisted of two plots of 1 ha each. One plot is for traditional monocultivar system and the other for the mixed-species. The plot with the traditional system was established more than 30 years ago and was based on the cultivar Orito (Musa acuminata AB type of genome) in La Maná site, and cultivar Barraganete (Musa balbisiana AA type genome) in El Carmen site. The other plot corresponded to a mixed Musaceae system based on 12 different Musa cultivars, planted in 2009 (Table 2). Each cultivar was represented by subplots of 24 plants, with borders of plants of the local

cultivar. Each subplot was repeated three times and randomly distributed in the ha. A spacing of $3 \times 3$ m (1111 plants ha$^{-1}$) was adopted in all plots.

**Table 2.** *Musa* ecotypes used in the mixed-species plot according to their genome.

| Banana: *Musa Acuminata* (AA) | Plantain: *Musa Balbisiana* (AB) |
| --- | --- |
| Orito (AA) | Barraganete (AAB) |
| Gros Michel (AAA) | Maqueño Verde (AAB) |
| Guineo Jardin (AAA) | Dominico (AAB) |
| Filipino (AAA) | Dominico Harton (AAB) |
| Williams (AAA) | Dominico Negro (AAB) |
| | Dominico Gigante (AAB) |
| | Limeño (AAB) |

### 2.3. Agronomic Management in the Experimental Plots

In both sites, the traditional systems were established by deep tillage practices, while tillage was minimum in the mixed systems. The main annual field operations are summarized as follows:

- Biweekly leaf pruning, eliminating folded senescent and dead leaves, as well as necrotic parts of leaves that have less than 30% necrosis.
- Shoots were eliminated every two months, selecting only one vigorous basal sucker as a replacement for the next generation.
- Peeling of the banana plant or elimination of dry leaf sheath from the pseudostem, every month during the rainy season and every two months in the dry season.
- Fruit harvesting every two weeks during rainy season and every three weeks for the dry season.
- Chime of corms from harvested plants.
- Fertilization (65 kg ha$^{-1}$ of N, 45 kg ha$^{-1}$ of P$_2$O$_5$ and 156 kg ha$^{-1}$ of K$_2$O) distributed at the beginning and end of the rainy season.
- Manual weed controls every month in the rainy season and every two months in the dry season, after evaluating the species present.
- Chemical weed control, two applications/year of glyphosate (2 L ha$^{-1}$) in the monocultivar system, one in the rainy season and another at the beginning of the dry season. The mixed plot received only one application during the rainy season. Throughout the dry season, manual weeding was performed every two months in both systems.
- Chemical control of *Cosmopolites sordidus* was applied once a year with 10 g plant$^{-1}$ of Alodrin RB only in the monocultivar system.

### 2.4. Arthropod Abundance and Identification

Arthropod samples were taken randomly from both (mixed and monocultivar) sites (La Maná and El Carmen) and two seasons of the year (rainy and dry). To collect the most representative taxes, two different trap systems were used: a pitfall trap consisting of a 1 L capacity vessel, buried at ground level during 72 h, to catch organisms falling into the container filled with water and liquid detergent. The second type of traps were "Chromatic" traps, which consists of yellow plastic plates (18 cm diameter) placed for 3 h on the floor, with a solution of water and liquid soap. A total of 80 samples were taken—40 for each site—and these, in turn, subdivided into 20 subsamples for each production system and 10 for each type of trap used. The collected individuals were taken to the university laboratory, where they were quantified and classified up to the taxonomic level of the order, dividing them by phylum, class (insect, arachnid, myriapod, mollusks and annelids) and orders. The indicators of biodiversity of individuals that constitute the soil macrobiota were determined for the total of arthropods found, which were described according to criteria of Moreno [14].

To determine the abundance of each arthropod taxon (order), two sampling periods were carried out, one between January and May during the rainy season and another between June and November during the dry season.

Arthropods were identified using taxonomic keys [15] or arriving at the level of orders and family and as far as possible; they were differentiated between beneficial and harmful specimens.

### 2.5. Statistical Analysis

Based on the total structure and number of the arthropod community collected, comparison of means of the abundance of orders or groups of arthropods were performed and a multifactorial analyses of variance (ANOVA) was carried out with factors: site (El Carmen y La Mana), season (rainy and dry season) and type of cultural system (mixed and monocultivar). Calculations were done using the R Commander program [15]; values of each plot were previously transformed to $\sqrt{(x + 1)}$.

In order to determine the abundance, richness, diversity (Simpson indexes), similarity (Jaccard coefficient) and equitability (J) of the soil macro fauna by site and production system, the PAST statistical program [16] was used.

The Simpson diversity index was calculated as $\lambda = \Sigma pi^2$, where pi = proportional abundance of species i, that is, the number of individuals of species i divided by the total number of individuals in the sample during the time period considered. This index shows the probability that two individuals taken at random from a sample are of the same species. It is strongly influenced by the importance of the most dominant species [16].

To determine the relationships between factors and arthropod abundances, a Principal Component Analysis (PCA) was performed, in such a way that the distribution of the sites and agricultural systems was visualized both in the rainy and dry season, taking into account the characteristic uncorrelated environmental factors, which explain much of the original total variability.

## 3. Results and Discussion

The influence of the three factors (mixed and monocultivar agricultural systems; season and sites) on the abundance of different arthropod orders are shown in Table 3. The insect orders, Coleoptera and Diptera, were significantly influenced by the agricultural system, while site was a determinant factor for the orders Hymenoptera, and Prostigmata ($p > 0.01$). Orders Collembola, Hemiptera and Orthoptera were represented in all factors. The season of the year had highly significant influence on the majority of the orders found. There were no significant differences between the two cropping systems and the localities for the orders Araneae, Diptera, Hymenoptera, and Spirobolida. These orders have great adaptation capacity to diverse environments. The long life-cycles of some of these species could be responsible for their presence throughout the year [17]. In addition, some species influence the transformation of biodegradable waste, especially the organic matter deposited on the soil surface, incorporating it into the edaphic system, through the tunnels and channels that the coleoptera excavate. This facilitates infiltration and aeration of the soil [18].

**Table 3.** Influence of agricultural system, season and site on the abundance of arthropod populations. The values result from the factorial analysis of variance when comparing these factors. Values were transformed to $\sqrt{(x + 1)}$ before the analysis.

| Order | Factor | Square Mean | F | *p* |
|-------|--------|-------------|---|-----|
| Araneae | Site | 0.09 | 1.40 | 0.240 |
| | Season | 11.05 | 168.96 | 0.000 |
| | Agricultural system | 0.01 | 0.17 | 0.686 |
| | Site × Season | 0.00 | 0.06 | 0.804 |
| | Site × Agricultural system | 0.08 | 1.19 | 0.280 |
| | Season × Agricultural system | 0.22 | 3.39 | 0.070 |

**Table 3.** *Cont.*

| Order | Factor | Square Mean | F | *p* |
|---|---|---|---|---|
| Coleoptera | Site | 0.62 | 12.79 | 0.001 |
| | Season | 4.36 | 89.16 | 0.000 |
| | Agricultural system | 0.67 | 13.74 | 0.000 |
| | Site × Season | 0.09 | 1.85 | 0.178 |
| | Site × Agricultural system | 0.30 | 6.22 | 0.015 |
| | Season × Agricultural system | 1.07 | 21.89 | 0.000 |
| Collembola | Site | 576.04 | 247.12 | 0.000 |
| | Season | 0.74 | 0.32 | 0.575 |
| | Agricultural system | 103.40 | 44.36 | 0.000 |
| | Site × Season | 5.35 | 2.30 | 0.134 |
| | Site × Agricultural system | 12.92 | 5.54 | 0.021 |
| | Season × Agricultural system | 2.97 | 1.27 | 0.263 |
| Diptera | Site | 0.72 | 1.64 | 0.203 |
| | Season | 14.77 | 33.72 | 0.000 |
| | Agricultural system | 4.76 | 10.87 | 0.002 |
| | Site × Season | 010 | 0.24 | 0.629 |
| | Site × Agricultural system | 0.12 | 0.28 | 0.601 |
| | Season × Agricultural system | 0.18 | 0.41 | 0.526 |
| Hemiptera | Site | 7.07 | 49.38 | 0.000 |
| | Season | 0.52 | 3.60 | 0.062 |
| | Agricultural system | 3.18 | 22.24 | 0.000 |
| | Site × Season | 3.76 | 26.25 | 0.000 |
| | Site × Agricultural system | 0.62 | 4.33 | 0.041 |
| | Season × Agricultural system | 0.21 | 1.48 | 0.227 |
| Hymenoptera | Site | 16.07 | 11.16 | 0.001 |
| | Season | 183.74 | 127.62 | 0.000 |
| | Agricultural system | 5.20 | 3.61 | 0.061 |
| | Site × Season | 33.49 | 23.26 | 0.000 |
| | Site × Agricultural system | 1.85 | 1.28 | 0.261 |
| | Season × Agricultural system | 3.13 | 2.17 | 0.145 |
| Orthoptera | Site | 8.06 | 29.27 | 0.000 |
| | Season | 2.19 | 18.81 | 0.000 |
| | Agricultural system | 3.03 | 26.05 | 0.000 |
| | Site × Season | 0.05 | 0.43 | 0.512 |
| | Site × Agricultural system | 0.59 | 5.10 | 0.027 |
| | Season × Agricultural system | 0.57 | 4.93 | 0.030 |
| Prostigmata | Site | 0.08 | 8.10 | 0.006 |
| | Season | 0.68 | 72.90 | 0.000 |
| | Agricultural system | 0.01 | 0.90 | 0.346 |
| | Site × Season | 0.08 | 8.10 | 0.006 |
| | Site × Agricultural system | 0.41 | 44.10 | 0.000 |
| | Season × Agricultural system | 0.01 | 0.90 | 0.346 |
| Spirobolida | Site | 0.00 | 0.12 | 0.730 |
| | Season | 0.36 | 20.28 | 0.000 |
| | Agricultural system | 0.05 | 3.00 | 0.088 |
| | Site × Season | 0.00 | 0.12 | 0.730 |
| | Site × Agricultural system | 0.05 | 3.00 | 0.088 |
| | Season × Agricultural system | 0.05 | 3.00 | 0.088 |

Arthropods from our field captures were distributed in three classes and eight orders (Table 4). In La Maná, we found that 56% of individuals belonged to order Collembola and 32% to Hymenoptera, while in El Carmen, 61.5% of them were classified into the order Hymenoptera. The least represented group were from the order Spirobolida and Prostigmata, with less than 0.5% values at both sites.

**Table 4.** Arthropod abundance distributed by taxonomic groups during the rainy and dry seasons in two ecological sites (El Carmen and La Maná) in 2014.

| Class | Order | La Maná | | | | El Carmen | | | |
|---|---|---|---|---|---|---|---|---|---|
| | | Rainy Season | % | Dry Season | % | Rainy Season | % | Dry Season | % |
| Arachnida | Arachnida | 0 | 0 | 44 | 1.9 | 4 | 1.4 | 48 | 3.3 |
| | Prostigmata | 0 | 0 | 6 | 0.3 | 0 | 0 | 12 | 0.8 |
| Hexapoda | Coleoptera | 22 | 1 | 58 | 2.5 | 17 | 5.7 | 40 | 2.7 |
| | Collembola | 1321 | 67 | 1038 | 44.9 | 47 | 15.9 | 98 | 6.7 |
| | Diptera | 40 | 2 | 107 | 4.6 | 48 | 16.3 | 158 | 10.8 |
| | Hemiptera | 40 | 2 | 93 | 4.0 | 30 | 10.2 | 13 | 1.0 |
| | Hymenoptera | 502 | 26 | 894 | 38.6 | 148 | 50.2 | 1063 | 72.8 |
| | Orthoptera | 40 | 2 | 67 | 2.9 | 1 | 0.3 | 22 | 1.5 |
| Diplopoda | Spirobolida | 0 | 0 | 7 | 0.3 | 0 | 0 | 6 | 0.4 |
| | Total | 1965 | 100 | 2314 | 100 | 295 | 100 | 1460 | 100 |

Irrespectively of the site and season factors, the type of agricultural system factor seems to influence the presence of orders Coleoptera, Collembola, Hemiptera and Orthoptera and project as good indicators of the differences between monocultivar vs. mixed systems. Differences were found between ecosystems: La Maná had more abundance in the quantity of specimens than El Carmen. La Maná is an area with higher relative humidity which indicates that this habitat presents more favorable conditions for the development and conservation of these insect groups, making this agroecosystem more stable and diverse.

Our results on species abundance are in accordance with [19], who affirmed that the collembola play an important functional role in the decomposition processes of dead plant matter, the nutrient cycle, and help in the formation of soil characteristics. As observed in agroecosystems from other latitudes [20,21], Collembola abundance can be related to climatic and edaphic factors, availability of nutrients and biodiversity. Collembola seems to represent the diversity and abundance of species in the agroecosystems, since there was a direct relation between their abundance and the edaphic humidity [22–24]. Thus, in La Maná, the abundance in Collembola indicates that there would be sufficient environmental humidity, even in the dry season.

The largest number of individuals (2686) was obtained in the mixed system from La Maná (Table 5), while the amount of arthropods captured in El Carmen was quite similar in both production systems, and in total, had 40% fold inferior to that of La Maná. The difference between sites can be due to the site characteristics where higher precipitation, temperature, relative humidity and the different soil texture, and a higher presence of soil organic matter measured in La Maná. Furthermore, both the stability of the system (5 years vs. 30 years) and the application of agrochemicals higher in the monocultivar than in mixed systems may account for the observed differences. Our results are in concordance with those of Lavelle et al. [8], who found that arthropod in soils in humid tropics, as El Carmen, are inferior in abundance because they must be adapted to habitats where micro-climate fluctuations can be very strong and usually have compact soils with low oxygen concentration and high brightness, and with few open spaces and low availability and quality of food.

La Maná presented an index of 0.59, indicating that this site has the characteristics suitable for the habitat of different organisms, thus, being a balanced ecosystem. In contrast, in El Carmen an index of 0.49 was obtained, being this ecosystem largely unbalanced and with less favorable conditions for the development of many arthropods (Table 6).

According to data from the Simpson (1-D) diversity indexes, it can be observed that the highest diversity of arthropods is found in soils from mixed farming systems. The agricultural system influences the equity (e ^ H/S). The number of arthropods was higher in the mixed system plots than in the monocultivar ones (Figure 1). On the other hand, agricultural systems and sites influenced the

dominance (D) and equity (J) of arthropods. The dominance index was higher in monocultivar systems, while the equity index was higher in mixed systems. To compare different agricultural systems (mixed versus monocultivar), several authors have used the richness of families. Wickramasinghe et al. [19] compared family richness among 24 pairs of mixed and monocultivar farming systems in Great Britain. They observed higher abundance and richness of insect species in mixed systems than in soils with single cultivars.

**Table 5.** Number of individuals registered in each type of production system at both sampling sites in 2014.

| Site | Production System | N° Individuals | Percentage % |
|---|---|---|---|
| La Maná | Mixed | 2686 | 44.5 |
| | Monocultivar | 1593 | 26.4 |
| El Carmen | Mixed | 936 | 15.5 |
| | Monocultivar | 819 | 13.6 |
| Total | | 6034 | 100 |

**Table 6.** The Simpson diversity index for each taxonomic group at each site in 2014.

| | La Maná | | El Carmen | |
|---|---|---|---|---|
| Order | N° Individuals | Simpson | N° Individuals | Simpson |
| Arachnida | 52 | 0.00015 | 44 | 0.00063 |
| Coleoptera | 80 | 0.00035 | 57 | 0.00106 |
| Collembola | 2359 | 0.30280 | 145 | 0.00689 |
| Díptera | 147 | 0.00118 | 206 | 0.01390 |
| Hemíptera | 133 | 0.00096 | 43 | 0.00061 |
| Hymenoptera | 1396 | 0.10604 | 1211 | 0.48051 |
| Orthoptera | 107 | 0.00062 | 23 | 0.00017 |
| Prostigmata | 6 | 0.00000 | 12 | 0.00005 |
| Spirobolida | 7 | 0.00000 | 6 | 0.00001 |
| Total | 4287 | 0.41209 | 1747 | 0.50384 |
| | 1-D | 0.58791 | 1-D | 0.49616 |

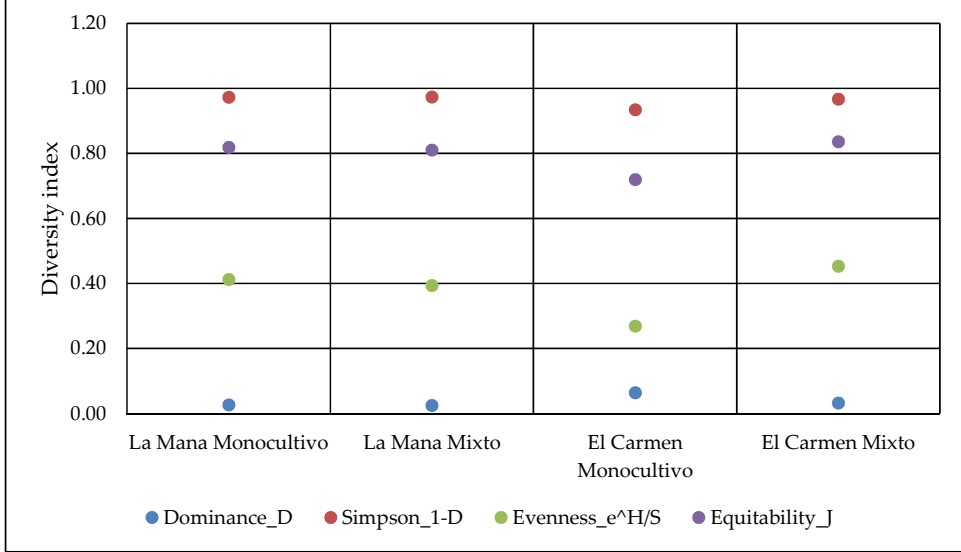

**Figure 1.** Simpson diversity index (1-D) and its three components; Dominance (D), Evenness (eˆH/S) and Equitability (J) in function of the agroecosystem and locality, considering the amount of arthropods found.

According to the Jaccard coefficient, the highest similarity based on arthropod families was found between the mixed and the monocultivar systems at both localities, with 80% (Table 7). At the level of orders, a closer similarity of 100% between monocultivar (La Maná) and mixed (El Carmen) systems was observed. Mixed systems (La Maná) and monocultivar (El Carmen) presented low percentage of similarity (Table 8).

**Table 7.** Jaccard index of similarity (%) based on different arthropod families between sites and production systems in 2014.

| Site/Agricultural System | El Carmen, Mixed | La Maná, Monocultivar | La Maná, Mixed |
|---|---|---|---|
| El Carmen, monocultivar | 0.80 (80%) | 0.55 (55%) | 0.42 (42%) |
| El Carmen, Mixed | – | 0.57 (57%) | 0.52 (52%) |
| La Maná, monocultivar | – | – | 0.79 (79%) |

**Table 8.** Jaccard index of similarity (%) based on different arthropod orders between sites and production systems in 2014.

| Site/Agricultural System | El Carmen, Mixed | La Maná, Monocultivar | La Maná, Mixed |
|---|---|---|---|
| El Carmen, Monocultivar | 0.88 (88%) | 0.88 (88%) | 0.56 (56%) |
| El Carmen, Mixed | – | 1.00 (100%) | 0.88 (88%) |
| La Maná, Monocultivar | – | – | 0.88 (88%) |

Among the observed arthropods, there were families categorized as predators or parasitoids of pests. These are highly beneficial for the crops. In banana, parasitoids are important biological control agents that regulate several pests, mainly diverse lepidopteran defoliators [20–23]. Braconidae was one of the most common families with beneficial insects. These parasitoid wasps are recognized as being endo or ectoparasitoids with idiobiont strategy exclusively attacking Lepidoptera, Coleoptera and *Diptera* during different stages of development. The tiny wasps of the Diapriidae family were commonly found in wet microhabitats and shady areas, such as La Maná ecosystem. They are mainly endoparasitoids and primary predators on larvae and pupae of a wide range of insects, especially flies (Diptera) [24,25]. Several studies in different polyculture models have shown that the composition and diversity of plants in or around the production systems have an influence on the number of insects (including parasitoids). The reason is that the diversity of plants attracts natural enemies by offering resources such as micro-habitat and food that, specifically on parasitoids, can be determinant for their species richness, their longevity and their level of parasitism [26,27]. The parasitic Hymenoptera are also good indicators of pesticide impacts, because they are more sensitive to pesticides than most other insects, including their host species [28,29].

Diapriids were the only parasitoids more abundant and with a higher species richness in conventional banana than in the mixture plot [30]. In our study, Braconidae and Diapriids (Table 9) were more abundant in mixed systems with moderate agrochemical inputs than in monocultivar with intensive agrochemical applications in both sites. Taking into consideration all the orders found independent of the agroecosystem, La Maná had three times more beneficial arthropods than El Carmen.

The principal component analysis (PCA) revealed scarce dispersion of the points for sites orders agricultural systems (Figure 2). This could be related to the relatively high number of specific orders favoring the characterization of ecosystems. PCA showed higher dispersion and fewer arthropods in the rainy season than the dry season, where the largest number of arthropods was concentrated. A greater variability among the orders observed in the different localities and agricultural systems were found in both seasons of the year (rainy and dry). In La Maná, Hemiptera and Collembola were closer related to the dry season and the mixed agricultural system, whereas Coleoptera and Orthoptera were more related to dry season and monocultivar production system. In El Carmen site, Hymenoptera, Arachnida and Diptera are more abundant in the dry season and mixed production

system and Prostigmata in the dry season and monocultivar agroecosystem. These two axes account for 55.0% of the total variability observed (Figure 2).

**Table 9.** Number of beneficial arthropods found in the different agroecosystems and sites.

| Order | Family | La Maná | | El Carmen | |
|---|---|---|---|---|---|
| | | Mixed | Monocultivar | Mixed | Monocultivar |
| Coleoptera | Coccinelidae | 2 | 1 | 4 | 1 |
| | Escaleridae | 0 | 1 | 0 | 0 |
| | Staphylinidae | 10 | 6 | 1 | 0 |
| Diptera | Asilidae | 5 | 8 | 1 | 2 |
| | Dolichopodidae | 28 | 15 | 37 | 6 |
| Hymenoptera | Braconidae | 37 | 23 | 5 | 2 |
| | Diapriidae | 120 | 81 | 36 | 19 |
| | Encyrtidae | 11 | 11 | 2 | 3 |
| | Myridae | 5 | 7 | 0 | 0 |
| | Sphecidae | 0 | 2 | 0 | 0 |
| **Total** | | 218 | 155 | 86 | 33 |

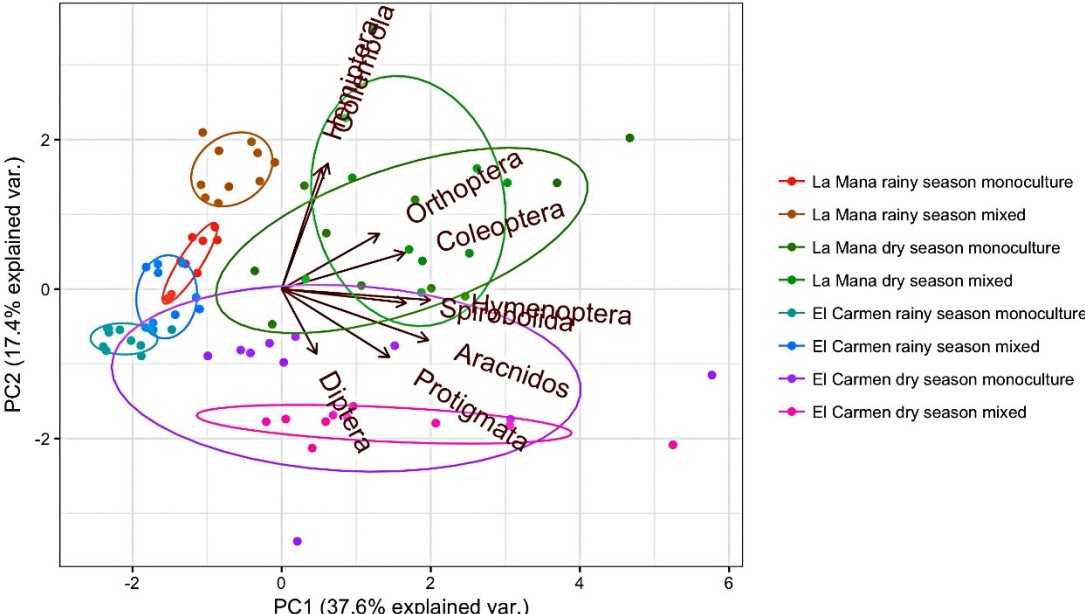

**Figure 2.** Principal component analysis (biplot) between sites, season, agricultural systems and number of arthropod orders found.

One of the most important aspects derived from these results are that a more diverse system presents a greater diversity of associated arthropods, depending on scale and context. This diversity is closely linked to the climatic conditions of the site, and the diversity of plants in the ecosystem [31], and it seems that the intraspecific diversity used in this study is enough to fulfil the needs of a diverse community of arthropods. Haddad et al. [32] demonstrated that the identity of plant species determines the abundance of arthropods, as well as affects different ecological processes within agricultural agroecosystems. The most relevant processes are biomass accumulation, decomposition rates and floor moisture. Thus, the results of arthropod diversity can be more influenced by the effects of the composition of plants than by the effects of the number of species [32,33]. The study by Rzanny et al. [33] was conducted in a temperate region, where winters induce an annual collapse in the abundance of all taxa. Contrastingly, a tropical climate is relatively stable and supports a relatively less perturbed community. We argue that the relatively stable conditions in the mixed *Musa*-based agroecosystems of

the current study (these systems were at least five years old) allowed a stable plant community over years, and the presence of a perennial crop reinforces this stability. Another factor that contributes to the establishment of a relatively stable arthropod community in this mixed Musaceae agroecosystem is the low to moderate application of pesticide treatments.

## 4. Conclusions

Abundance analysis of the macro edaphic fauna and identification of the beneficial families allowed the determination of the equilibrium level of the Musaceae agroecosystem. The mixed type of production system provides plant diversity, which favors arthropod abundance and permits lower agrochemical application without yield penalties in comparison to the monocultivar. Within Hexapoda, the orders that presented larger populations were Collembola and Hymenoptera, based on the abundance and distribution they presented. The order Hymenoptera dominated in all the treatments, both by its abundance and by its distribution in the studied localities, even in ecosystems with ecological imbalance. Consistent with results from temperate studies, the mixed Musaceae production system was the one with the greatest presence of soil macro fauna, with the order Collembola being the most diverse, which gives us the guideline to say that this order is associated with the quality of the soils.

The management practices in agroecosystems can alter the community structure of pests' natural enemies, which can consequently influence their biocontrol. Since the functional composition of natural enemy communities, rather than taxonomic diversity, drive pest suppression efficiency, it is necessary to employ the functional approach to investigate the impact of management on natural enemies.

Our findings show that intraspecific diversity could be a good option to include in an IPM strategy for small and medium farmers and may help in the design of Musaceae agroecosystems to enhance the ecological regulation of pest management, without putting on the farmer the constraint of management different crops. Further research should explore the effect of combinations of various cultural intraspecific diversity systems with a more detailed study of the arthropod community present, down to at least genus identification in order to better define biodiversity indicators, especially considering that agricultural biodiversity will be essential to cope with the predicted impacts of climate change, and to detect more resilient farm ecosystems.

**Author Contributions:** Conceptualization, D.V.-A. and C.S.-C., methodology, D.V.-A., C.S.-C., M.L., C.P. and P.D.-S.; formal analysis, D.V.-A. and M.C.-G.; investigation, D.V.-A. and C.S.-C.; writing—original draft preparation, D.V.-A., C.S.-C., M.L., C.P. and P.D.S.; writing—review and editing, D.V.-A. funding acquisition, C.S.-C. All authors have read and agreed to the published version of the manuscript.

**Funding:** This research was funded by Bioversity International Grant contract number: (I-R_1362).

**Acknowledgments:** We are very grateful to Bioversity International Rome Italy and Technical State University of Quevedo, Ecuador for their financial and scientific support of this work. I sincerely thank the Universitat Autònoma de Barcelona (UAB), Spain for their contribution in this study.

**Conflicts of Interest:** The authors declare no conflict of interest.

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
