# Peer review of "Arthropod Diversity Influenced by Two Musa-Based Agroecosystems in Ecuador"

_agriculture, doi:10.3390/agriculture10060235_

Round 1

Reviewer 1 Report

Overall, interesting research, however I have several concerns:

  1. Authors mention that the mixed plantation receives lower inputs compared to monoculture, is this something they have documented or is it only an assumption. More detail into what "lower inputs" means would be beneficial.
  2. Authors do not identify much difference among the 4 treatments (two types of plantations in two localities), they mention they might not be as different, however these results might be because they only focused on order and didn't go beyond for the analyses. I would HIGHLY recommend that the authors identify their species further, preferably beyond family. 
  3. It wasn't clear, but based on the methodology it seems there was only 1 timepoint in the dry season and 1 in the rainy season. That seems too low, especially if the collection methods last 1 week tops. It does not seem to be representative enough of the whole season. Multiple timepoints are highly encouraged.
  4. Finally, the authors mentioned how Hymenoptera were the most abundant orders in both sites, however, they do not seem to elaborate on what these Hymenoptera are. The only results I found where they elaborate a little is on table 9 where they present some families of beneficial, however these only account for 297 out of 1396 and 67 out of 1211 individuals for La Mana and El Carmen respectively.

Author Response

  1. In materials and methods, I have added and explained all the agronomic tasks and the inputs that two agricultural systems receive. The agricultural management carried out in the two systems does not have many differences.
  2. The special situation of this research is that there is very little information and work of this type in the region and that it was wanted to take advantage of what the Bioversity International project did by comparing two banana production systems managed in a very similar way and whose main difference was monoculture vs. intraspecific diversity of mixed systems and the importance of intraspecific diversity to improve biodiversity indicators through the presence of arthropods at the order level, nematodes and soil microbial load.
  3. Two types of traps and two samplings were used in each season (rainy and dry), where the traps were placed randomly, having representative information in each season.
  4. Arthropods associated with the cultivation of Muses are not commonly found beneficial families that can regulate some type of pest in the culture. The most economically important pest is Cosmopolites sordidus and there are very few predators for this pest, noting that among the larvae and egg predators, the Coleoptera Hololepta sp. and Alegoria dilatata. Some studies also indicate that ants of the genus Camponotus sp can be found frequently in some plantations, preying on black weevil larvae. Other predators are the "tijeretas" (Dermaptera: Forficulidae)

Reviewer 2 Report

This is nice and interesting study bringing new insights to diversity of arhtropods in Musa agroecosystems in tropical regions of South America. I recommend publication after some minor changes:
1) Even I am not native speaker, I had feeling that english is bad in some parts of MS, for example some sentences are with grammar mistakes etc, including typing errors. I would recommend reading by native speaker which enhance the english validity of MS.

2) Line 45-47. There are more studies concerning importance of natural enemies, management and pest presence. Add the reference by Michalko and Košulič 2020: https://www.tandfonline.com/doi/full/10.1080/09670874.2019.1601292?casa_token=yjg7iYKXwwMAAAAA%3Aeqrtb2meZTJ-TZ5FD4feg-hTsGBkfZ7xxFB80w0Mvee_QYTWehOwHqfDAzHTxg5mgh_NaF_KAp3Y

3) Line 52-53, There are more interesting examples inlcuding effect of agriculture and forestry on diversity in ecosystems, such as Kosulic et al. 2016: https://journals.plos.org/plosone/article?id=10.1371/journal.pone.0148585

4) Authors should describe better the management informations in Musa intensive stand - what pesticide, how often, to see how it can affect the natural enemies.

5) Statistical analyses are described in Material and Methods very briefly, I suggest to describe in in more details.

Author Response

1. A general revision of the manuscript was made with an expert where the grammatical and writing part were improved.

2 - 3. In points 2 and 3, the indicated references that the reviewer recommends are added to the manuscript. Musaceae-based farming systems have found several natural enemies of Cosmopolites sordidus the main pest, pointing out that among the larvae and egg predators, the Coleoptera Hololepta sp. and Alegoria dilatata. It is also indicated that the ants of the genus Camponotus sp Other predators are the “tijeretas” (Dermaptera: Forficulidae)

4. In materials and methods, I have added and explained all the agronomic tasks and the inputs that two agricultural systems receive. The agricultural management carried out in the two systems does not have many differences.

5. Within the statistical analysis that was used: factorial ANOVA, PCA, arthropod biological diversity indices and Jacard coefficient for the similarity of the factors, they are universal general analyzes that many researchers know, that is why a summary of those used and in this way I justify this point

Round 2

Reviewer 1 Report

Authors did not make major changes to the paper with respect to the suggested methodological concerns. 

Author Response

All the changes suggested by the referees and the editor have been made, which is evidenced in the manuscript in the form of change control in the text. Important and relevant concepts have been incorporated in the introduction part of several studies carried out by Michalko and Košulič and irrelevant text has been removed in this same section.
In the Materials and methods section, a restructuring has been carried out both in the agronomic management and in the statistical part, which has expanded the explanation with more detail of the statistical technique used.
In the conclusions section, two more paragraphs were added detailing the importance of the results for the scientific community and the farming community, giving a regional and international scope of the impacts generated in the management of the different agricultural systems and climatic conditions.
All the changes made to the manuscript in order to improve its scientific quality are evidenced in the correctly identified text.
Best regards